# Phage Encounters Recorded in CRISPR Arrays in the Genus *Oenococcus*

**DOI:** 10.3390/v15010015

**Published:** 2022-12-20

**Authors:** Yasma Barchi, Cécile Philippe, Amel Chaïb, Florencia Oviedo-Hernandez, Olivier Claisse, Claire Le Marrec

**Affiliations:** UMR Oenologie 1366, Univ. Bordeaux, INRAE, Bordeaux INP, Bordeaux Sciences Agro, Institut des Sciences de la Vigne et du Vin, 33882 Villenave d’Ornon, France

**Keywords:** fermented beverage, *Oenococcus*, bacteriophage, CRISPR, spacers, host spectra

## Abstract

The *Oenococcus* genus comprises four recognized species, and members have been found in different types of beverages, including wine, kefir, cider and kombucha. In this work, we implemented two complementary strategies to assess whether oenococcal hosts of different species and habitats were connected through their bacteriophages. First, we investigated the diversity of CRISPR-Cas systems using a genome-mining approach, and CRISPR-endowed strains were identified in three species. A census of the spacers from the four identified CRISPR-Cas loci showed that each spacer space was mostly dominated by species-specific sequences. Yet, we characterized a limited records of potentially recent and also ancient infections between *O. kitaharae* and *O. sicerae* and phages of *O. oeni*, suggesting that some related phages have interacted in diverse ways with their *Oenococcus* hosts over evolutionary time. Second, phage-host interaction analyses were performed experimentally with a diversified panel of phages and strains. None of the tested phages could infect strains across the species barrier. Yet, some infections occurred between phages and hosts from distinct beverages in the *O. oeni* species.

## 1. Introduction

Grapes are currently the world’s largest fruit crop. In 2020, the world area under vines, corresponding to the total surface area planted with vines for all purposes (wine, juices, table grapes and raisins) was estimated at 7.3 Mha [1]. Large parts of the collected fruits are turning to grape wines, which are among the most iconic fermented products worldwide. From an ecological point of view, fermenting wine is a complex and fluctuating environment, which is characterized by successional trajectories of distinct communities of microorganisms, which often represent a small part of the resident grape microbiota [2,3]. In most red and dry white wines, the process starts with the selection of the fruit and its fermentation into alcohol by yeasts (AF), followed by a malolactic fermentation (MLF), by which L-malic acid is decarboxylated to L-lactic acid. MLF serves several purposes: to reduce the harsh acidity of malic acid, give a concomitant modest increase in pH as well as increase wine microbial stability. This essential step to obtain high quality grape wines is commonly driven by the lactic acid bacterium (LAB) *Oenococcus oeni*. Formerly known as *Leuconostoc oenos*, the species was assigned to a new genus in 1991 and has long been the only known representative species [4]. It was later shown to occur naturally in other beverages such as ciders [5] and kombucha teas [6,7]. More recently, various inventories have expanded our knowledge of the horizons of the habitat of *O. oeni* to other niches such as water kefirs and brine-type fermented foods [8], as well as fructose-rich environments including rotting fruits, fruit juices such as mango juice [9], coffee pulp [10] and beebread [11]. LAB from such fructose-rich niches have gained much attention in recent years and have a promising role in food fermentations [12]. Sister species have been also progressively identified: *O. alcoholitolerans* from sugarcane fermentation vats of Brazilian distilleries producing bioethanol and Cachaça [13], *O. kitaharae* from composting residues of shochu distillates [14], kombuchas and water kefir [8], and *O. sicerae* from ciders and water kefirs [15,16]. Yet, relatively little is currently known about sister species relationships in the *Oenococcus* genus, and the evolutionary and ecological drivers that control the populations in their different and sometimes overlapping niches (water kefirs, ciders, kombuchas) need a better understanding for the further optimization of fermented-food production. Mobile genetic elements (MGE) such as plasmids, genetic islands and bacteriophages may be candidates for such forces. In particular, bacteriophages may profoundly influence their hosts through phage predation and phage-mediated horizontal transfer [17]. The real oenococcal phage diversity has yet to be fully appreciated. So far, only phages of *O. oeni* have attracted attention during winemaking. The bacterium interacts with a diversity of temperate [18,19], ex-temperate [20] and strictly lytic [21,22] bacteriophages in musts and wines (red, dry and sweet white wines, sparkling wines), which may lead to delayed or stuck MLF. All oenophages described so far have a tailed morphology with a dsDNA genome and belong to the *Caudovirales* order [23]. More than 95% of the known oenophages are temperate phages [24] and carry out site-specific recombination (SSR). The two major constituents of the SSR unit, corresponding to the phage integrases and attachment sites, have been studied in detail and this resulted in the clustering of prophages into six major groups (Int_A_ to Int_F_), which are related to the integration site in the host chromosome [19]. The reconstructed phylogeny of all sequenced oenophages showed that the sequences segregated into three distinct and well-supported clusters. Cluster 1 comprises all temperate oenophages, except members of the Int_D_ group, which belong to cluster 2 and have been recently assigned to the new genus *Sozzivirus* [23]. Cluster 2 also harbours the strictly lytic Vinitor phages, which are related to a prophage in the gut-associated LAB *Convivina intestini* [22].

Strains of *O. oeni* can defend themselves against viral infections with diverse antiphage mechanisms, such as restriction-modification and superinfection exclusion systems, which have been identified by whole genome sequencing [24]. Yet, CRISPR-Cas adaptive defence mechanisms [25] are lacking conserved homologs in the sequenced strains of *O. oeni* [24,26]. CRISPR-Cas systems are unevenly distributed across taxa and environments, and recent analyses have shown that less than half of mesophilic bacteria have CRISPR arrays [27,28]. More importantly, bacteria with high mutation and slow growth rates, such as *O. oeni* [26,28], are much less likely to evolve CRISPR-based immunity [29,30,31]. The situation in the sister species, namely *O. sicerae*, *O. kitaharae,* and *O. alcoholitolerans* stands in sharp contrast to that of *O. oeni,* and the presence of CRISPR-Cas loci has been documented in strains of all three species [4,26,32]. Even though no detailed exploration of these loci has been conducted, their presence raises interesting questions regarding the conditions for establishing and maintaining CRISPR-mediated adaptive immunity amongst oenococci.

The CRISPR-Cas system is composed of clustered, regularly interspaced, short repetitive elements (direct repeats) separated by unique sequences (spacers) matching foreign DNA and an operon of *cas* genes that encode for proteins that process the crRNA and cleave DNA targeted by the spacers. Spacer acquisition is the first step of the CRISPR-Cas mechanism, named ‘adaptation’ or ‘immunisation’ [27]. Importantly, sequence identity between the spacers and cognate protospacers (on the phages genomes or other MGE) is required for immunity [25]. CRISPR arrays therefore contain a record of previous encounters with phages in the sequence of their spacers. In addition, position in the array and the number of nucleotide mismatches with host protospacers hint at the elapsed evolutionary time since a phage and its host last interacted. Therefore, spacer sequences may provide a historical perspective on foreign DNA exposure by oenococcal hosts and thereby can be used as an indicator of their evolution [33,34].

As two more whole-genome sequences from kefir-associated strains (*O. sicerae* and *O. kitaharae*) recently became available [16,35], we set out to analyse and compare the genomic sequences of the CRISPR elements (repeats, spacers and Cas proteins) among oenococci, offering a powerful tool for investigating phage–host interactions. Our data will provide new insights to the evolution and impact of phage specificity within complex bacterial communities associated to fermentations.

## 2. Material and Methods

### 2.1. Bacteria and Culture Conditions

Six strains were obtained from the Oenological Biological Resource Center (CRBO, ISVV, Villenave d’Ornon, France). They include four *O. oeni* strains, including IOEBS277 (red wine), CRBO1381 and CBO1384 (cider), and BL4 (kombucha), and two *O. kitaharae* strains associated with water kefir (CRBO2176) and Shochu (NRIC0649).

Strain *O. alcoholitolerans* JP736 was kindly provided by Prof. Marcos Morais (Federal University of Pernambuco, Brazil), and *O. sicerae* UCMA15228 was purchased from the DSMZ Culture Collection.

All strains were grown in MRS broth adjusted to pH 4.8 (Man Rogosa Sharpe) (Difco, Fischer Bioblock Scientific, Illkirch, France) at 25 °C. When requested, agar 1.6% (wt/vol) was added. All media were heat or filter-sterilized.

### 2.2. Phage Lysates Used in the Study

Five previously characterized oenophages were used in this study. OE33PA is an ex-temperate phage that harbours an Int_B_-type integrase sequence [18,20]. Vinitor (162, 27 and 28) and Krappator 27 are strictly lytic phages [21,22]. They were isolated from wine samples and propagated on strain *O. oeni* IOEBS277 to obtain high titre lysates, which were stored at 4 °C, as previously described [18]. Strain IOEBS277 does not contain endogenous phages and is sensitive to all oenophages so far isolated in our laboratory [19]. The accession numbers of the four phages are given in Appendix A.

A set of 30 partially characterized temperate oenophages were collected from wine samples during recent surveys [21,36]. They were available as phage lysates and stored at 4 °C.

### 2.3. PCR Identification of of Lysogenic Strains

Assessment of lysogeny in three unsequenced bacterial strains, BL4, CRBO14221 and CRBO14223, was obtained as follows. Bacterial DNA was prepared and used as the template in six PCR tests that distinguish the Int_A_, Int_B_, Int_C_, Int_D_ [18,21], Int_E_ and Int_F_ groups [36] among oenophages, based on their integrase (int) sequence. A Biorad i-Cycler was used for the amplification reactions, which were achieved in a 25 µL volume using the Bio-Rad Taq PCRMaster Mix kit and 0.2 mmol/L of each primer. All oligonucleotides were purchased from Eurofins MWG-Operon (Munich, Germany) and are available in Appendix A.

### 2.4. Mitomycin C Induction of Lysogenic Strains

Phage induction by mitomycin C (MC) was examined in lysogenic strains of *O. oeni* (CRBO14221, CRBO14223, and BL4), *O. sicerae* UCMA15228 and *O. alcoholitolerans* JP736. Cells were grown until an OD600 of 0.2 and added with MC 1 µg/mL for *O. oeni* and *O. alcoholitolerans*, and 0.5 µg/mL for *O. sicerae*. Following lysis, the cultures were cleared of cellular debris by centrifugation (30 min, 5000× *g*, 4 °C). The supernatant was then filtered through a 0.2 μm pore size sterile filter and stored at 4 °C.

### 2.5. Host Spectrum Determination

Ten-fold serial dilutions of the phage lysates were prepared in phage buffer (50 mM TRIS HCl pH 7.5; 0.1 M NaCl; 8 mM MgSO_4_). The phages were enumerated on indicator strains using the MD/SP (Multiple Dilutions on Single Plates) method [37] on MRSΦ agar [18]. Tested strains were IOEBS277, CRBO1381, CRBO1384 and BL4 (*O. oeni*), CRBO2176 and DSM17330 (*O. kitaharae*), UCMA15228 (*O. sicerae*) and JP736 (*O. alcoholitolerans*). Bacteria were grown to exponential phase, and 200 μL of the bacterial culture was mixed in 5 mL of top MRSΦ agar and poured onto MRSΦ agar plates. Spot assays were performed using 8 μL of the purified phage and the diluted samples. Incubations were carried out under anaerobic conditions (AnaeroGen; Oxoid) at 25 °C for 3–7 days to allow plaque formation. The presence of individual plaques was recorded as a positive test.

When phage lysates only revealed clear lysis in the first two dilution spots, with no individual plaques, nor any sign of lytic activity in further dilution spots, the lysate was dialyzed (Spectra/Por^®^ Float-A-Lyzer^®^ G2 dialysis tubing MCWO 100 kDa) against MRS broth (3 × 4 h) and subsequently against phage buffer (1 × 4 h). Pure and ten-fold diluted samples were tested using a DAL (Double Agar Layer) method. A volume of 100 µL of phage sample was mixed with the bacterial culture in soft agar and poured on MRSΦ agar. In the DAL method, the phage particles proliferate in the soft agar, while bacteria are fed from the underlying solid agar [37].

### 2.6. Genome Sequencing

Purification of the prophages from strains CRBO142221 and CRBO14223 was done using MC-induced cultures (50 mL). Phage OE33PA6 was propagated on strain IOEBS277 at a multiplicity of infection of 0.01.

The three phage lysates were centrifuged (30 min, 5000× *g*, 4 °C), and the supernatents were filtered. The phage particles were concentrated by ultracentrifugation, and double-stranded DNA was extracted as described previously [38]. Whole-genome sequencing was performed at the Genome-Transcriptome facility of Bordeaux. DNA libraries were prepared using the Nextera XT DNA library preparation kit (Illumina, San Diego, CA, USA). Genomic DNA was sequenced using an Illumina MiSeq using 2 × 250 bp paired-end libraries. Reads were assembled using SPAdes [39] with default parameters (read correction and assembler). The genome sequences of the three phages were submitted to GenBank, and their accession numbers are given in Appendix A.

### 2.7. Bioinformatic Analysis

The complete genome of 187 prophages was recently retrieved from 134 GenBank genomes [19]. Most analysed lysogens (*n* = 86) harboured a single prophage, and 48 poly-lysogens contained two (*n* = 43) or three (*n* = 5) prophages. Strains were collected worldwide from different fermented beverages (red, dry and sweet white wines, sparkling wines, and more recently, cider and kombucha tea). We also included the two single prophages found in CRBO14221 and CRBO14223. All accession numbers and tRNA insertion sites where prophages are integrated in the chromosome of lysogens are given in Appendix A. This constitutes our main data set of 189 prophages.

Predicted functional categories were associated with all available phage genes using the RAST pipeline [40]. The in silico translated protein sequences were used as queries to search for sequence homologs in the non-redundant protein database at the National Centre for Biotechnology (including the viral genome database). Deduced proteins were searched for function using BLAST v2.10.0 and a cutoff E value of 0.001. Searches for distant homologs were performed using HHpred [41] against different protein databases, including PFAM (Database of Protein Families), PDB (Protein Data Bank), CDD (Conserved Domains Database) and COG (Clusters of Orthologous Groups), which are accessible via the HHpred website.

Phage genome comparisons were conducted using the Genome-BLAST Distance Phylogeny (GBDP) method using VICTOR [42] under settings recommended for prokaryotic viruses [43] at: http://ggdc.dsmz.de/phylogeny-service.php (accessed on 6 October 2022). The resulting intergenomic distances were used to infer a balanced minimum evolution tree with branch support via FASTME, including SPR postprocessing for each of the formulas D0, D4 and D6. Branch support was inferred from 100 pseudo-bootstrap replicates each. Trees were rooted at the midpoint and visualized with FigTree as already described [22].

### 2.8. Typing System for Rapid Assesment of Diversity amongst Uncharactized Oenophages

The phylogeny of temperate, ex-temperate and strictly virulent phages showed (i) the presence of similar prophages in distinct strains, (ii) the extensive modularity of phage genomes and (iii) the segregation of phages into distinct and well-supported clusters [19,24]. Taking as input the set of 189 available prophage genomes, as well as the four newly sequenced oenophages OE33PA and Vinitor 162, 27 and 28 [21,22], we identified diverse genetic loci, whose presence or absence could rapidly group the so far uncharacterized oenophages isolated during previous surveys.

We focused on five main categories, in which we targeted six unique and essential functions: lysogeny establishment and maintenance (Integrase, int and LexA- or Cro/CI-type repressor proteins, LM) [19], replication (replisome initiator protein, Rep), head morphogenesis (Small Terminase unit, TerS) and encapsidation (Tape Measure Protein, TMP) [36]. Last, we retained a toxin (Doc), which was part of a putative toxin–antitoxin system occasionally localized in the moron module of a few phages, including that of OE33PA [20]. For each of the six targeted proteins (Int, LM, Rep, TerS, TMP, Doc), sequences from the 193 oenophage genomes were aligned using ClustalOmega at https://www.ebi.ac.uk (accessed on 20 September 2022). The multiple sequence alignments resulted in the identification of groups of conserved sequences in the amino acid sequences (>90% identity) in related phages. Visual examination of the alignments produced with ClustalOmega revealed blocks that contained invariant regions suitable for primer prediction using Primer3 [44]. The list of primers used in this study is given in Appendix A.

### 2.9. CRISPR-Cas Systems and Anti-CRISPR Identification

All data were analysed for the presence of CRISPRs and *cas* genes with the use of the CRISPRCasFinder program with default parameters [45], which enables the identification of direct repeat consensus boundaries and the extraction of related spacers and *cas* genes. A multi-FASTA file containing the unique spacers identified in each species (*O. alcoholitolerans*, *O. sicerae* and *O. kitaharae*) was used as a reference to blast a customised database of selected plasmids (*n* = 5) and phages (*n* = 34) of *O. oeni*. We performed a command-line NCBI-BLASTN, with 85% identity and 90% query coverage. Next, a BLAST search of the identified spacers against the original dataset of 193 oenophages was done and did not retrieve additional spacers. Each matching protospacer identified in a given phage was also manually compared to the most related phage genomes in the database, in order to identify putative additional mutations.

Maximum likelihood phylogenetic trees of Cas proteins were constructed using the PhyML algorithm at http://www.atgc-montpellier.fr/phyml/ [46], the latest version of which includes automatic selection of the best-fit substitution model for a given alignment. A Bayesian-like transformation of aLRT (aBayes), as implemented in PhyML, was used to estimate branch support.

Finally we analysed genomes of *O. ciserae* for the presence of anti-CRISPR (Acr) encoding genes. They often have an immediate downstream gene coding for a putative transcription regulator, named the Aca (Acr associated) protein. Such Acr-Aca loci were searched for in the genome of *O. ciserae* UCMA15558 using the Acr Database [47].

## 3. Results and Discussion

### 3.1. Distribution and Architecture of CRISPR Loci in the Genus

Whole-genome sequencing analysis of strains collected from wine, cider and kombucha have repeatedly confirmed the lack of CRISPR-Cas mechanisms in the *O. oeni* species [4,26,32]. In this study, we set out to better understand the native interspecies occurrence of CRISPR in three novel oenococcal species: *O. sicerae*, *O. kitaharae* and *O. alcoholitolerans*. To this aim, the genomes of five strains available from the NCBI GenBank Database (*O. alcoholitolerans* UFRJ-M7.2.18^T^, *O. sicerae* UCMA15228^T^ and OAL24, *O. kitaharae* DSM17330^T^ and CRBO2176) were analysed with CRISPRCasFinder. A single CRISPR array was identified in the chromosome of *O. alcoholitolerans* UFRJ-M7.2.18 (Figure 1). We also observed that plasmids possessed very questionable CRISPR in this strain, containing some direct repeats (DR) with no associated *cas* genes (Appendix A). These loci are likely reminiscent of degenerate and non-functionable systems. Consistent with a previous report [32], we successfully detected a single CRISPR system in *O. kitaharae* DSM17330^T^ and two arrays in *O. sicerae* UCMA15228^T^ (Figure 1). The coexistence of several CRISPR-Cas systems in the same genome has been previously described in several food LAB and bifidobacteria [48,49,50].

CRISPR-Cas systems are classified into six types and more than 30 subtypes based on phylogeny and use of different *cas* genes [27,28]. As summarized in Figure 1, arrays that contained the Cas9 endonuclease and Csn2 protein signatures of type II-A CRISPR loci were identified in *O. alcoholitolerans* UFRJ-M7.2.18, *O. kitaharae* DSM17330 and *O. sicerae* UCMA15228. These observations are consistent with recent findings that type II systems are widespread among LAB [27,48]. Top BLAST Hits (TBH) for the Cas9 proteins in the three tested strains were retrieved and used to construct a phylogenetic tree, which is provided in Appendix A. The sequence in *O. kitaharae* was more closely related to that of *O. alcoholitolerans* and had 47% identity with the Cas9 protein of *Leuc. pseudomesenteroides* (e-value 0). In contrast, the Cas9 protein from *O. sicerae* was closer to that of *Fructobacillus tropaeoli* (E value 0; 45% identity).

Interestingly, a second CRISPR-Cas system was found in *O. sicerae*. It was classified as a type I-E array and was larger than the type II system. This is due to the fact that type I uses multiple Cas protein complexes for interference, whereas type II systems only use Cas9 proteins [27]. The nuclease Cas3 is the hallmark protein of type I systems, and the TBH of the protein in *O. sicerae* was identified in *Lactiplantibacilli* (E value 0; 58% identity; Appendix A). The widespread occurrence and efficient use of type I-E CRISPR-Cas systems to edit various chromosomal loci in LAB have been recently documented [48].

Strikingly, CRISPR-Cas systems were absent in *O. sicerae* OAL24 and *O. kitaharae* CRBO2176, which were both isolated from water kefir. As illustrated in Figure 1, pairwise genomic comparisons of each strain with the corresponding type strain (*O. sicerae* UCMA15228 and *O. kitaharae* DSM17330) led to the identification of a signature consisting of a single DR, suggesting the deletion of the whole type II-A CRISPR arrays in both OAL24 and CRBO2176.

Rearrangements that are more complex are suggested in *O. kitaharae* CRBO2176 and are illustrated by the insertion of three extra open reading frames (Figure 1). A partial type I RM system was found with putative *hsdR* (endonuclease) and *hsdS* (specificity unit) genes with head-to head gene organization. No *hsdM* gene was found in the locus. The putative HsdS protein had 73% identity to similar systems in *Leuconostoc sp* C2 (CP002898.1) and *Leuc. kimchii* (CP037939.1), which were both isolated from homemade kimchi [51]. The HsdR protein was formed by two domains. The N-terminus (271 aa) had 85% identity to an HsdR protein in *O. sicerae* UCMA15228 (QAS70346), while the C-terminus (92 aa) had 99% identity to the C-terminal domain of a phage integrase (QAS70349) affiliated to the same bacterium. The third gene of interest in CRBO2176 was related to an Ig-like surface protein. HHPred analyses predicted the presence of a choline binding domain (C-terminal segment 120–270 aa) resembling a vaccinal antigen in *S. pneumoniae* (4CP6-A; prob. 99.36; E value 8 × 10^−10^). Our observations made in strain CRBO2176 suggest extensive recombination events within the CRISPR-Cas region in *O. kitaharae* and possible association of the system with prophages, as reported in other bacterial models [52]. In *O. sicerae* OAL24, a full deletion of the type I CRISPR-Cas systems occurred without leaving evidence of its past presence in the genome.

Each of the four confirmed CRISPR arrays in *O. alcoholitolerans*, *O. sicerae* and *O. kitaharae* consisted of several exact DR of different sizes, ranging from 29 to 36 nucleotides (Table 1). The whole DR sequences in *O. alcoholitolerans* and *O. kitaharae* were identical (Table 1), although such sequences are often species specific [27].

A total of 197 spacers was identified, and their full list is given in the Appendix A. We observed repeated blocks of spacers in *O. sicerae* UCMA15228, and to a lesser extent in *O. kitaharae* DSM17330, which possibly resulted from duplications [27]. Consequently, the number of unique spacers was slightly smaller in both species, and respective ratios of unique/total spacers of 0.89 and 0.98 were found. We also analysed the spacer repertoire relatedness between the four arrays and found a complete replenishing of the four loci with unique spacers.

Typical arrays usually contain fewer than 50 spacers in bacteria [53]. Accordingly, the repertoire associated with the type II systems in *O. alcoholitolerans*, *O. sicerae* and *O. kitaharae* contained 15, 23 and 57 unique spacers, respectively (Table 1).

Contrasting occupancy of the type I CRISPR locus was observed in *O. sicerae* UCMA15228 in that the array contained 102 spacers. As mentioned earlier [54], the number of spacers in an array is a result of a compromise between better protection offered against abundant, diverse and faster evolving viruses by a larger spacer repertoire and a higher physiological cost of maintaining a longer array.

Overall, the presence of 125 spacers in *O. sicerae* UCMA15528^T^ is intriguing and implies that the strain frequently comes under attacks by phages during the cider fermentation process. *Oenococcus* sp. and lactobacilli are the most common and dominant LAB identified in the bacterial communities of apple juice by-products [55]. Yet, Ledormand and co-authors recently failed to isolate phages infecting *Oenococcus sp*. from a set of 120 samples (cider, apple must, crushed apples), and a single strictly lytic phage infecting *Liquorilactobacillus mali* was isolated [56]. Moreover, deep-sequenced DNA viromes of ciders did not lead to the recovery of viral sequences related to oenophages [57]. An early collection of the cider samples that took place before the onset of MLF may explain the obtained results.

As shown in Table 1, the type strains of *O. alcoholitolerans*, *O. sicerae* and *O. kitaharae* harbour four chromosomal CRISPR loci, which account for a total of 183 unique spacers (Appendix A). Based on the signature genes and gene arrangements, two types (I-A and II-E) were identified. Despite limited availability of bacterial genomes in each species, our study (i) confirms an uneven distribution of CRISPR-Cas immune systems in the *Oenococcus* genus, with the *O. oeni* species devoid of such viral defence systems, and (ii) also suggests within-species variability. The later observation has also been reported in *S. thermophilus* and Lactobacilli [27,34,48]. Additional studies are now needed to better understand the variation in the existence, number and functionality of CRISPR-Cas systems in oenococci. An interesting question will be to assess whether fluxes in the function, acquisition by horizontal gene transfer and deletion of CRISPR-Cas loci occur as a strategy to cope with different habitats, especially when phages are a major source of mortality.

### 3.2. Selection of Representative MGE of O. oeni for Interspecies CRISPR Targeting

Our objective was to generate a set of representative and non-redundant viral and plasmid sequences to be used for interspecies CRISPR targeting. We first made a comparative analysis of the full genome sequences of oenophages to identify phage cluster relationships and focused on a limited number of phage genes and their deduced proteins. As a second step, the analysis provided a rapid typing system to screen for substantial novelty and diversity in a collection of 30 uncharacterized phages that have been collected over the past years in our laboratory [18,21,36].

The relationship between the integrase and TMP sequences and phage clustering has been proposed for wine associated oenophages [18,19,36]. In this study, we expanded our system to additional proteins (LM, Rep, TerS and Doc) to improve the phage-typing discriminatory power of our scheme. To this aim, comparisons of whole genome sequences were performed. They included 189 prophages, Vinitor phages (162, 27 and 28) and OE33PA (Appendix A). This analysis revealed that proteins encoded by divergent genes could perform analogous functions, depending on the phage (Figure 2). Hence, we identified a set of seven distinct classes of proteins associated with a TMP function (TMP 1–7), providing a direct view of the modular and combinatorial nature of oenophage genomes. Six types of integrase (A to F) and immunity repressor (LM 1–6) and seven types of terminase (TerS 1–7) were also identified. Replication modules had five distinct organisations. Each was represented by a specific protein specifying either a RecA-like ATPase (Rep1), an helicase (Rep2), a DUF1351 domain-containing protein (Rep3), an excinuclease (Rep4) or a phage replisome organiser (Rep5). Last, we retained a toxin (Doc), which was part of a putative toxin-antitoxin system occasionally localized in the moron module of a few phages, including that of the Int_A_ prophage of strain IOEB0608.

From our in silico analysis, each phage genome was associated with a barcode corresponding to its cognate TMP, TerS, integrase, LM, Rep and presence/absence of Doc proteins. Our queries identified 27 distinct patterns among the 193 phages analysed (Appendix A). A representative phage for each pattern was selected for further analysis. In a few cases, two or three phages sharing the same pattern were included in our study, as they originated from different steps and/or wine types, yielding a final set of 33 phages.

For each of the 32 conserved phage proteins of the scheme (6 Int, 6 LM, 5 Rep, 7 TerS, 7 TMP, 1 Doc), all available nucleotide sequences were compared in order to design primers. They were used for screening our collection of poorly characterized phages. OE33PA6 was the only phage showing an original combination of sequences (Int_A_, LM1, Rep1, TerS2, TMP1, Doc^-^) compared to the 27 identified patterns. The genome of phage OE33PA6 was sequenced (Appendix A) and added to our set of oenophage genomic data, which accounted for 34 members. Of note, the limited novelty in the analysed set of uncharacterized oenophages confirms that free replicating particles in wines are mostly temperate oenophages, resulting from excision of the prophage genome out of the chromosome of lysogens [36].

Additional data to input to the database were plasmids, which are frequently targeted by CRISPR arrays. Five sequences in *O. oeni* were included in our queries: pOeni1 and pOeni2, which replicate using a rolling-circle mechanism [58], pRSE2 and pRSE3, which use a theta mode of replication [59], and a large plasmid recently found in strain AWRIB429 [60].

### 3.3. Intra- and Interspecies CRISPR Targeting of Known Prophages and Plasmids of Oenococcus oeni

To evaluate the protection by CRISPR-Cas systems against the foreigner nucleic sequences, we performed a command-line NCBI-BLASTn with 85% identity and 90% query coverage against our local database of *O. oeni* phages (*n* = 34) and plasmids (*n* = 5) with the 183 unique spacer sequences extracted from the CRISPR arrays found in *O. alcoholitolerans*, *O. sicerae* and *O. kitaharae*. Eleven spacers were found as protospacers in viral and plasmid sequences of *O. oeni*, consistent with the CRISPR-Cas function as a protective immune system against acquiring foreign nucleic sequences (Table 2). None of these identified spacers had 100% identity to the targeted MGE in *O. oeni*, and one to four distinct mismatches were observed between spacers and protospacers (Table 2). A single protospacer was present in plasmids pOeni1, pOeni2 and pAWRIB429.

Importantly, we assessed whether CRISPR may also target the backbone of the chromosome of *O. oeni*. We analysed the genome data of 33 lysogens harbouring the prophages that were retained in our set of oenophages. No additional targeting was identified along the entire length of their chromosome, including the regions harbouring remnant phages [19]. Similar data were obtained with non-lysogenic strains of *O. oeni* [19].

The CRISPR-Cas system in *O. alcoholitolerans* showed no spacer matching with *O. oeni* phages or plasmids. All eleven spacers matching *O. oeni* phage or plasmid protospacers were identified in *O. kitaharae* (SP_OKII_2, SP_OKII_9 and SP_OKII_15) and *O. sicerae* (SP_OSII_1, SP_OSII_21, SP_OSI_5, SPO_SI_7, SP_OSI_43, SP_OSI_46, SP_OSI_78 and SP_OSI_89) (Table 2). Their abundances were inversely correlated with phylogenetic distances between *O. oeni* and its sister species [15].

Many spacers were found in middle and leader distal positions in each of the three arrays (Table 2). Yet, five spacers (SP_OKII_2, SP_OKII_9, SP_OSII_1, SP_OSI_5, SPO_SI_7) were shown to locate in the leader proximal positions in their respective array. As documented in the past, spacers integrated into the 5′ end are more recently acquired compared to those at the 3′ end [33]. In *O. sicerae*, the fraction of detected protospacers was higher for the type I than for the type II CRISPR-Cas system. In addition, there may be a priority among oenococcal strains to select the type II CRISPR system against virulent phages as well as plasmids.

The findings that ten spacers in *O. kitaharae* and *O. sicerae* specifically targeted phages of *O. oeni* may reflect movement between the species and supports various host–phage interactions in the *Oenococcus* genus. Yet, the majority of the spacers in the three identified arrays remain uncharacterized and correspond to the so-called « CRISPR dark matter » in *O. kitaharae* and *O. sicerae*. We next investigated whether these congeneric species that inhabit diverse environmental settings were also connected to other LAB species through bacteriophages. To this aim, we looked for matches to genomes from different microbial taxa in the Genbank databases. We found that the repertoires associated with type II-CRISPR arrays lacked detectable homologues in the GeneBank nucleotide database. Yet, a single spacer in the type I array in *O. sicerae* (SP_OSI_32, 5′ TTTCATTTAAATATGCATGCTGATGATGACGG 3′) matched various phage sequences, with one or two mismatches. All protospacer-containing genes encode a phage replisome organizer protein. The matches include a bacteriophage sp. isolate named ctsqZ15a, associated with human metagenomes [61], as well as a variety of prophages in *Lactiplantibacillus plantarum* (most lysogenic strains originated from kimchi, dairy products, beer and healthy infant faecal samples) and *Liquorilactobacillus mali* (strain LM596 from apple juice from cider press) (Figure 3, Appendix A).

We conclude that the high prevalence of orphan spacers in *O. kitaharae* and *O. sicerae* implies the existence of a large number of undiscovered phages, plasmids, ICE or other sequences in the bulk of the genetic landscape of MGE among oenococcal species, which remains to be discovered. This would also suggest that local phages that most frequently infect the tested strains of *O. kitaharae* and *O. sicerae* in their respective environment (Shochu and apple cider, respectively) would be phylogenetically distinct from the currently characterized wine-associated oenophages. Alternately, the limited number of spacer matches could be attributed to the presence of substantial plasmids or phages that constantly evolve, resulting in them escaping from the mechanism.

### 3.4. Autoimmunity in O. sicerae

One of the potentially detrimental costs associated with CRISPR-Cas viral defence is autoimmunity, which arises when CRISPR arrays acquire spacers that target the host genome; this is often thought to be lethal [27]. BLASTn comparisons of spacers from *O. kitaharae* DSM17330 and *O. sicerae* UCMA15228 against their respective whole genome sequences revealed five additional self-targeting spacers (STS) in *O. sicerae* UCMA15228 and none in *O. kitaharae* (Table 2).

All STS in *O. sicerae* UCMA15228 were located in the type I CRISPR array (SP_OSI_30, SP_OSI_36, SP_OSI_86, SP_OSI_87 and SP_OSI_98) and had 100% sequence identity to a 45.3 kb prophage integrated in a 16-bp sequence (5′ACTCCTGTTCGGGGCA3′) at the 3′ end of a tRNA^Leu^ gene (Appendix A). The culture was observed to lyse in the presence of mitomycin C 0.5 µg/mL, suggesting the ability of the prophage to excise and produce bacteriophage particles. Interestingly, the vicinity of the locus used for site-specific recombination in *O. sicerae* is similar to that of the *attB_B_* used by Int_B_ prophages in *O. oeni* [19]. Salient features of the identified prophage are the presence of genes specifying a tRNA and a putative potassium voltage-gated channel subfamily KQT (ORF4) (Figure 4). A BLASTp search against non-viral organisms revealed a putative K^+^ channel protein from various *O. oeni* and *O. kitaharae* strains as the most similar proteins (E values e−115 and e-100, 89 and 88% aa identity in *O. kitaharae* DSM17330 and *O. oeni* CRBO1395, respectively). The corresponding regions are not assumed to be mobile in these bacteria [19].

The presence of STS in *O. sicerae* is consistent with recent data showing that prophages are associated with extensive CRISPR-Cas autoimmunity [62,63,64]. A model was recently proposed where primed adaptation in type I systems amplifies STS by acquisition of new spacers from both prophage and prophage-adjacent regions [62]. Yet, targeting of the prophage region did not lead to STS of endogenous flanking genomic regions in *O. sicerae*. As seen in Figure 4, the prophage genes targeted by CRISPR in *O. sicerae* included genes encoding the integrase (ORF1), a putative transglycosylase soluble domain-containing protein that may be a minor tail component (ORF47), and the tail associated lysin (Tal) baseplate protein (ORF50) (Figure 4). The later gene was targeted by three distinct STS, which were part of the oldest spacers in the CRISPR array, suggesting these were the initial hits.

As part of the ongoing bacterial-phage arms race, escape from the lethal outcome of auto-immunity occurs through various mechanisms in bacteria, including the acquisition of mutations on the target sequence and/or inactivation of CRISPR–Cas functionality via, for example, mutation or deletion of essential sequences in the array [62,63,64]. Yet, we predicted the 5′-TTA-3′ sequence as the conserved protospacer adjacent motif (PAM) upstream of the five STS (as well as in three of the five protospacers in oenophages). We therefore investigated the presence of anti-CRISPR (Acr) proteins, which also play a major role in auto-immunity evasion and in the co-evolution of phages and their bacterial hosts [65]. We performed bioinformatic searches for Acr orthologs in the prophage of *O. sicerae* UCMA15228 using the online database AcrDB [47] and retrieved four candidate proteins with high confidence. They are encoded by ORF2, ORF3, ORF7 and ORF8 (Figure 4). Products of ORF1 and ORF7 were assigned to AcrIIA1 and AcrIF11, respectively. Acr candidates did not have neighbouring Aca homologs. It was recently observed that only a small percentage (23% in bacteria and 8.2% in viruses) of Acr and Aca form operons and are difficult to predict due to their genomic and structural diversity [66].

### 3.5. Protospacer-Containing Prophages and Genes in O. oeni

Since host CRISPRs in *O. kitaharae* and *O. sicerae* bear evidence of a few encounters with oenophages, we next investigated how the protospacers matching the ten identified spacers (SP_OKII_2, SP_OKII_9, SP_OSII_1, SP_OSII_21, SP_OSI_5, SP_OSI_7, SP_OSI_43, SP_OSI_46, SP_OSI_78 and SP_OSI_89) were distributed in our representative set of 34 phages of *O. oeni*. To better capture the diversity among oenophages, a phylogenetic tree using the full genome of the selected phages was constructed, and all results are provided in Figure 3.

Approximately 65% (22/34) of the representative oenophages contained at least one protospacer matching a bacterial spacer sequence in *O. kitaharae* or *O. sicerae* (Figure 3). Matches were distributed across the two main phylogenetic clusters, earlier described as I and II [19].

Comparing the spacers to known oenophages also revealed differential targeting of related viral groups by CRISPRs, with possible distributed immunity to some phylogenetically related oenophages belonging to subcluster I.1 (Figure 3). Of note, the targeted oenophages from cluster I.1 were recently shown to dominate the viral population upon completion of the MLF in red wines [30]. In contrast, phages affiliated with subclusters I.2 and II.2 (strictly lytic Vinitor phages) [22] had noticeably fewer protospacers, while phages in cluster II.1 contained no protospacers. The latter cluster includes the Int_D_ temperate phages, which were recently assigned to the new genus *Sozzivirus* [23]. Such phages may have a more restricted niche and less probability to interact with *O. sicerae* and *O. kitaharae*. This hypothesis needs further investigation.

Most spacers could match more than one virus in cluster I.1. We observed that prophages AWRIB847_ProB_, AWRIB202_ProB_, AWRIB150_ProB_ and IOEB0501_ProA_ had the most protospacers (five), followed by AWRIB508_ProA_, CRBO14223_ProA_, S161_ProA_, L26.1_ProB_, CRBO11105_ProB_ (four), IOEBB10_ProB_ and AWRIB127_ProC_ (three) (Figure 3). Three of these prophages, namely IOEB0501_ProA_, S161_ProA_ and L26.1_ProB_, were previously shown to be released as free particles upon mitomycin induction of their lysogenic hosts and to form plaques on various strains of *O. oeni* [19].

We next analysed the targeted-gene characteristics in oenophages, and some spacers shared a region of homology with the genes encoding tail components (TMP, Dit, Tal), capsid protein (head closure knob protein) and DNA packaging machinery (portal protein), which play vital roles in the phage life cycle (Table 2). It is worth noting that the protospacer-containing gene *tmp2* in S161_ProA_ was used in our initial phage-typing scheme.

We also observed that the *dit* gene was targeted twice by the type II system from *O. kitaharae*. Similarly, three spacers from the *O. sicerae* type I-E system (SP_OSI_43, SP_OSI_78, SP_OSI_89) matched gene regions encoding hypothetical proteins that were located in the predicted replication module of the phage genomes (Table 2). The immunity offered by this type of CRISPR-Cas system could therefore prevent phage replication via the destruction of these critical components and enhance the viability of bacteria in phage-rich environments. The protospacers incorporated into the *O. sicerae* type I CRISPR locus as SP_OSI_78 and SP_OSI_89 were of particular interest due to their high frequency among phages from cluster I (15 of 27 phages). The sequences were both located in a gene specifying a conserved putative bi-functional primase-polymerase (97–100% identity at the protein level among the 15 phages) (Table 2, Figure 3). Orthologous protein sequences are also found in other phages such as the dairy phages PLE2 (51% identity, YP_009282384) and 98201 (46% identity, YP_009286328) infecting *Lactobacillus casei* and *Lactococcus lactis*, respectively. This ORF was lacking among the 12 remaining oenophages from cluster I (Figure 3). Several variants of the two protospacers were found, and sequence alignments of SPOSI78, SPOSI89 and the corresponding protospacers amongst oenophages are given in Figure 5A. Yet, mutations were not specific to the targeted portions of the gene and accounted for a total of 20 positions, which were distributed along the ORF. We propose that both protospacers incorporated as SP_OSI_78 and SP_OSI_89 are ancestral and may represent a common origin for phages of *O. oeni* and *O. sicerae.* This may reflect exposure to a virus from the same viral family that has undergone sequence divergence over evolutionary times.

More recent spacers were also found and may arise from contemporary phages that may be present in the different habitats of the congeneric oenococcal species. In particular, SP_OSII_1 was incorporated as a spacer from both strictly lytic phages Vinitor 27 and Vinitor 23, but not Vinitor 162. The phage gene that was targeted in Vinitor 23 corresponded to the phage replisome organizer (Rep5). Analysis of the Rep5 protein sequences from Vinitor 23 and its homologue in Vinitor 162 is given in Figure 5B,C and reflects various genome rearrangements. One of them in Vinitor162 eliminated the protospacer matching with SP_OSII_1. Future work is now needed (i) to assess whether such variations are a response of phages to escape CRISPR pressure and (ii) to isolate additional strictly lytic phages of *O. oeni* and *O. sicerae*.

### 3.6. Search for Cross-Infections in the Oenococcus Genus

The presence of a few recent spacers in the CRISPR loci of strains of *O. sicerae* and *O. kitaharae* with homology to predicted oenophage genomes is suggesting that some phages may not be strictly species-specific and may infect a broader range of strains in the *Oenococcus* genus. Although phage selection is generally considered to be narrow for the host, increasing evidence suggests that phages have a broad host range in nature, such as recently shown for Staphylococcus phages [67]. Furthermore, CRISPR spacers are now powerful datasets used to predict bacterial hosts of MGE [25]. We thus sought to assess the host-range of reference oenophages and performed preliminary interactions studies using the double-agar assay between phages and oenococcal hosts from different species and beverages (Table 3).

We tested seven phage lysates. The set included three phages from cluster I: the ex-temperate OE33PA (cluster I.1) propagated on strain IOEBS277, the Int_E_ phage from strain CRBO1384 (cluster I.2) and the Int_F_ phage from strain BL4 (cluster I.2) (Figure 3). The two latter strains have a single prophage as assessed by PCR and MC-induction assays. Both phage lysates were obtained upon MC-induction of cultures grown in MRS broth. Vinitor 162 was the representative of cluster II (cluster II.2). We also tested the Krappator 27 phage. No integrase was amplified by PCR, suggesting that this phage is strictly lytic. In addition, PCR assays targeting the Vinitor-associated sequences in *rep5*, *ters7* and *tmp6* were also negative, suggesting that Krappator 27 may represent a new phage group amongst oenophages. Last, MC-induced cultures of *O. sicerae* UCMA15228 and *O. alcoholitolerans* JP736 were also tested. For the latter strain, analysis of phage DNA from a MC-induced culture suggested the presence of a single prophage (Appendix A).

The seven phage lysates were tested against eight strains belonging to the *O. oeni*, *O. sicerae*, *O. kitaharae* and *O. alcoholitolerans* species, which were collected from distinct beverages. Four phages infecting *O. oeni* (OE33PA, Krappator 27, Vinitor 162 and the Int_E_ phage from strain CRBO1384) formed plaques on at least one host. Yet, their host-ranges did not extend beyond the *O. oeni* species. Interestingly, although we could not observe any cross species viral infections, positive infection assays were detected across beverage (wine, cider and kombucha) in the *O. oeni* species.

## 4. Conclusive Remarks

The application of next generation sequencing to comparative genomics has enabled in-depth characterization of genetic variation between strains of *Oenococcus oeni* obtained from wine, apple cider and kombucha. Regarding the strategy used to fight invaders, all strains lack a CRISPR mechanism, regardless of the beverage and fruits from which they originated. In contrast, our data suggest that CRISPR-Cas systems are present in *O. sicerae, O. kitaharae* and *O. alcoholitolerans*, even though we currently ignore the extent to which such systems are operational in the three sister species, warranting further analyses. Of the two genomes available in each of the *O. sicerae* and *O. kitaharae* species, only one contains a CRISPR-locus, suggesting that the presence of CRISPR-Cas systems may be strain- rather than species-dependent. The ecological factors that drive variation in CRISPR-Cas prevalence across natural microbial communities associated with beverages remain unclear. Our preliminary data obtained in *O. alcoholitolerans* are invaluable as a baseline study of whether plasmids may facilitate their dissemination and how ecosystems could favour loss of CRISPR systems in certain conditions in the *Oenococcus* genus. The interplay and redundancy between different phage defence systems may also contribute to CRISPR-Cas distributions in different environments and should be investigated as more genomes become available in sister species.

CRISPR-Cas spacers represent a sequential quelling of former intruder encounters, and the retained spacers reflect evolutionary phylogeny or relatedness of the strains. In the present study, we observed that the CRISPR-Cas system in *O. sicerae* and *O. kitaharae* showed spacer matching with *O. oeni* phages and plasmids. This may indicate that members of these bacterial species have acquired immunity against related phages in their specific environments. Yet, with the performed phage–host range experiments, we could not observe any cross-species infection between phages and strains. The only positive results were obtained within the *O. oeni* species. Experimental efforts are needed to isolate more strains, as this will help assessing whether phage adsorption on host cell surface relies on specific features in *O. oeni* as compared to the three sister species.

## Figures and Tables

**Figure 1 viruses-15-00015-f001:**
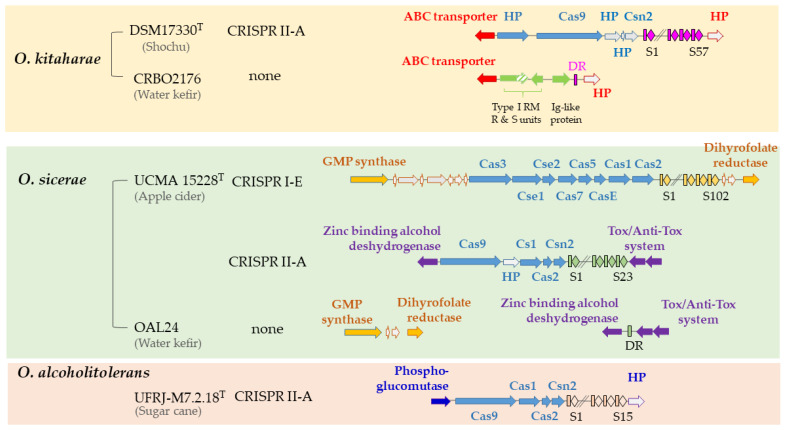
Distribution and architecture of CRISPR loci in the *Oenococcus* genus. Direct repeats (DR) and spacers are represented by rectangles and lozenges. Genes coding for hypothetical proteins (HP) are in grey. Double slash marks represent DNA regions that are not shown. Sequence ID for the restriction endonuclease R and S subunits and putative Ig-like proteins in *O. kitaharae* CRBO2176 are MCV3295975.1, MCV3295974.1 and MCV3295973.1, respectively.

**Figure 2 viruses-15-00015-f002:**
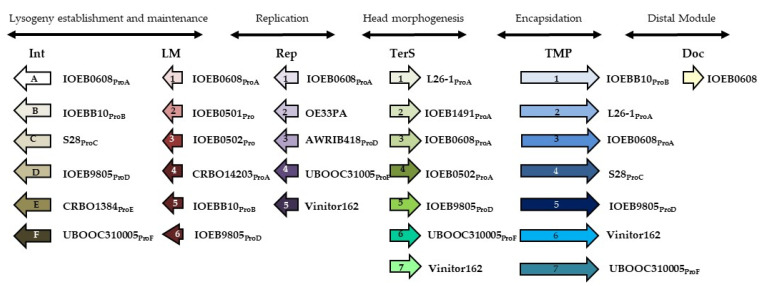
Scheme for the classification and rapid genotyping of oenophages. LM, lysogeny maintenance; Rep, replication-associated protein; TerS, small unit of the terminase; TMP, tape measure protein; Doc, toxin–antitoxin system. Representative phages containing the genes encoding the distinct proteins are indicated. Pro indicates that the phage is a prophage. Because some strains are poly-lysogens, the integrase type (A to F) is also indicated.

**Figure 3 viruses-15-00015-f003:**
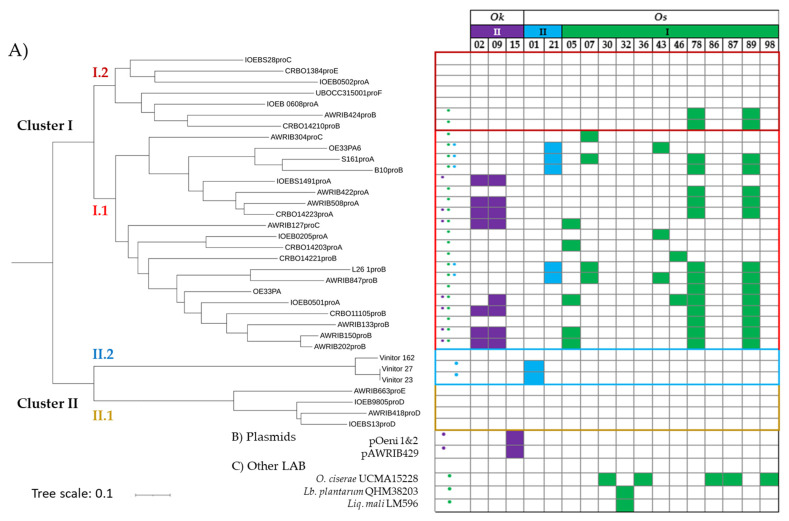
Distribution of the protospacers in oenophages. (**A**) Phylogenetic tree of the 34 phages of *O. oeni*, including OE333PA (ex-temperate), OE33PA6 (temperate) and Vinitor (strictly lytic), which were isolated as free replicating phage particles from must or wine samples. Other genomes correspond to prophages, and reference to their cognate integrase group was also added, since many lysogens harbour two or three prophages belonging to distinct integrase groups. In the leftward column, dots were used to summarize the presence of protospacers targeted by the CRISPR of *O. kitaharae* (purple) or *O. sicerae* (blue and/or green); (**B**) presence of spacers in plasmids of *O. oeni*; (**C**) presence of spacers in the genome of other LAB. *Lb. plantarum*, *Lactiplantibacillus plantarum*; *Liq. mali*, *Liquorilactobacillus mali*.

**Figure 4 viruses-15-00015-f004:**
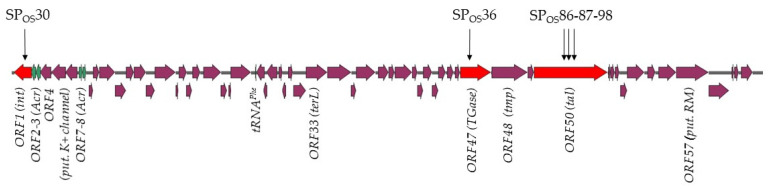
Auto-immunity and location of STS in *O. sicerae* UCMA15228. The genomic organization of the prophage genome targeted by the type I CRISPR system is shown. The location of the protospacers incorporated into the CRISPR locus as STS (SP_OSI_30, SP_OSI_36, SP_OSI_86, SP_OSI_87 and SP_OSI_98) are indicated above the map. The four putative *acr* genes are in green. Tgase: Transglycosylase.

**Figure 5 viruses-15-00015-f005:**
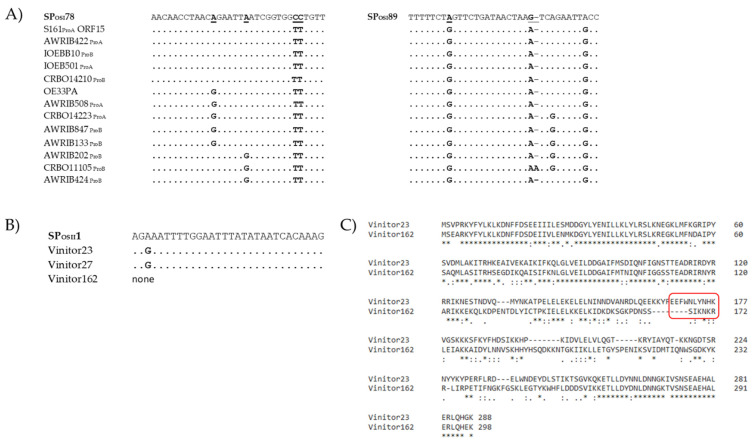
Variability of the protospacer sequences incorporated as SP_OSI_78, SP_OSI_89 (**A**) and SP_OSII_1 (**B**) in oenophages. Analysis of the Rep5 proteins in Vinitor 23 and its homologue in Vinitor 162 (**C**). The variable region marked in red results from the deletion of the protospacer matching SP_OSII_1 in Vinitor 162.

**Table 1 viruses-15-00015-t001:** Characteristics of CRISPR arrays in the *Oenococcus* genus.

CRISPR Loci		DR		Spacers			
Species	Type	Size (bp)	Sequence	Size (bp)	Total	Unique	With Matches in *O. oeni*
*O. kitaharae* DSM17330^T^	II-A	36	GCTTCAGATGTGTGTCAGATCAATGAGGTAGAACCC	30	57	56	3
*O. sicerae* UCMA15228^T^	II-A	36	GGGTGTCACCCCATTAATCTGACATACAACTGAAGC	29–31	23	21	2
	I-E	29	AGGATCACCCCCGCTTGTGCGGGGAATAC	32–33	102	91	5
*O. alcoholitolerans* UFRJ-M7.2.18^T^	II-A	35	GCTTCAGATGTGTGTCAGATCAATGAGGTAGAACC	30–31	15	15	0

**Table 2 viruses-15-00015-t002:** List of the unique protospacers found in MGE from *Oenococcus sp.* (phage or plasmid) and function of the genes targeted by CRISPR systems.

Type	Sequences of Spacers (SP), Protospacers and Relevant Phages	Phage/Plasmid Functions Targeted by CRISPR Systems
II *Ok*	SP_OKII_2IOEB1491_ProA_	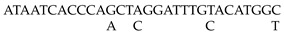	Dit(phage baseplate protein, WP_032824876)
SP_OKII_9IOEB1491_ProA_	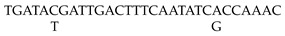
SP_OKII_15pOeni1/2; pAWRIB429	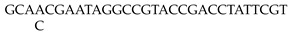	Intergenic sequence
II *Os*	SP_OSII_1Vinitor27	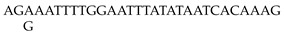	Rep5 (Replication inititation protein QNO11543)
SP_OSII_21S161_ProA_	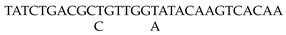	Tal(peptidase, KGH92654)
I *Os*	SP_OSI_5IOEB0501_ProA_	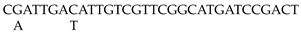	Head closure protein(WP_002820745)
SP_OSI_7S161_ProA_	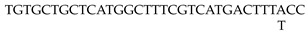	TMP2(KGH92652)
SP_OSI_43IOEB0205_ProA_	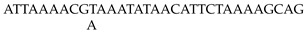	HP(WP_002825766)
SP_OSI_46IOEB0501_ProA_	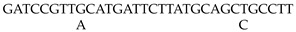	Portal protein(WP_032822098)
SP_OSI_78S161 _ProA_	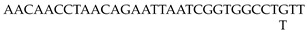	Bifunctional DNA primase/polymerase (WP_002824107)
SP_OSI_89S161_ProA_	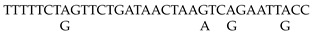
STS_OSI_30Resident prophage	AATGTGCTTGATTCAAGCTATGGCAATGACTC	Transglycosylase soluble domain-containing protein
STS_OSI_36Resident prophage	CCCACCAGCTGTCTGGCATTGAGATAGTTCGT	Tyrosine-type recombinase/integrase (QAS70227)
STS_OSI_86Resident prophage	ATTTATTAGGACGAGGGACTCCGATGGTGAAC	Tal(WP_243148548)
STS_OSI_87Resident prophage	ACCATCGTTTGATAAGTCGAGCTCTCAACTTT
STS_OSI_9Resident prophage	CAAGAAGTTCCACTAATCACTGTCGTCGCAGC

*Ok*, *O. kitaharae*; Os, *O. sicerae*; I and II refer to the types of CRISPR-Cas systems; SP, spacers; STS, self-targeting spacers; Dit, distal tail protein; Rep, replication-associated protein; Tal, tail-associated lysin; TMP, tape measure protein.

**Table 3 viruses-15-00015-t003:** Interactions between phages and members of the *Oenococcus* genus.

Phages				Bacteria, Species and Origin											
				*O. oeni*						*O. k*		*O. s*		*O. a*	
				W	C			Ko		Kef	S	C		E	
Host species and habitat		Name (type of lysate)	Clus-ter	IOEB S277	CRBO 1381	CRBO 1384		BL4		CRBO 2176	NRIC 0649	UCMA 15228		736	
*O. oeni*	W	OE33PA (P)	I.1	++	++	+		++		−	−	−		−	
		Krappator27 (P)	II.1	++	++	-		++		−	−	−		−	
		Vinitor 162 (P)	II.2	++	− *	+		− *		−	−	−		−	
	C	CRBO1384_ProE_ (MC)	I.2	++ **	−			−		−	−	−		−	

	Ko	BL4 _ProF_ (MC)	I.2	−	−	−				−	−	−		−	

*O. c*	C	UCMA15228 (MC)	nd	−	−	−		−		−	−			−	

*O. a*	E	*O. alcoholitolerans* JP736 (MC)	nd	−	−	−		−		−	−	−			


W, wine; C, cider; Ko, kombucha; E, ethanol distillates; Kef, water kefir; P, lysate obtained upon the lytic propagation of the phage on strain IOEB277; MC, lysate obtained upon the mitomycin C induction of the lysogen; the diagonal line refers to immunity to superinfection. Spot assays were performed using 8 μL of the purified phage and the diluted samples. +/++ indicate the relative number of plaques; * a halo was observed for both pure and ten-fold diluted phage lysate, while the 10^−2^ diluted sample did not yield any individual plaque; ** pinpoint plaques were observed and phage particles were unstable at 4 °C, suggesting that the resident prophage in CRBO1384 is a difficult-to-culture phage.

## Data Availability

All data used in this study is available from the Genbank database (https://www.ncbi.nlm.nih.gov/genbank/ (accessed on 6 October 2022)).

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
