# Peer review of "Phage Encounters Recorded in CRISPR Arrays in the Genus Oenococcus"

_viruses, 2022, doi:10.3390/v15010015_

Round 1
Reviewer 1 Report
Brief summary
This paper aims to identify connections of different oenococci through the bacteriophages and plasmids (MGE) interacting with this genus. By identifying and describing CRISPR-Cas systems in congeneric species of O. oeni the authors prove that diverse defence systems are indeed employed by Oenococcus sp., despite their lack in O. oeni. They screen identified CRISPR spacers against a customized dataset of O. oeni MGE and present in silico evidence that confirms cross-species interactions along the evolutionary time. They also describe inter- and intraspecific protospacer matches to the identified spacers and the protospacer distribution among MGE of O.oeni. After performing host-range experiments with strictly lytic and temperate phages of O. oeni and other congeneric species they find no positive in vivo evidence that confirms cross-species phage interactions. They attribute the negative in vivo results to the limited availability of oenococci isolated to date.
Broad comments on strengths and weaknesses of the study
This work by Barchi et al., has overall an excellent, on-the-spot experimental and innovative design and reports very interesting results that broaden our knowledge on past and present interactions among Oenococcus sp. and their cognate phages. I read the paper with great interest and found the results and conclusions important for the phage community and the whole work worth publishing, albeit only after some major improvements are been made.
Specifically, it is evident to me that the computational analyses are generally meticulously done, although some improvement of the methods and results presentation (figure and data availability) is necessary, to facilitate reading and interpretations (for further please read specific e.g. below and in the attached pdf). I would especially like to stress out the need for data availability here as this is to facilitate the reviewing process and validity of results in the next round, and ensure study reproducibility and credibility long-term.
More specific comments/remarks that should please be addressed:
I. Throughout the manuscript:
- I suggest that the authors replace the term “lytic” with “strictly lytic”; "lytic" is not a widely accepted term and can lead to confusions. For reasons why you the term “lytic” should be avoided in general see paper with doi: 10.1093/femsle/fnw047
- Please ensure that you always write out the names of the elements you mention instead of providing how many these are. An e.g. is in line 343 where you use a number (ten) instead of the names. Naming the elements you refer to makes the text more clear and easy to follow.
- Check carefully all your references and correct wrong citations or citations that are not relevant or are erroneously placed (e.g. in line 255, the references given do not support that a single protospacer was present in these plasmids; also in line 55 the reference is not relevant).
II. Introduction
The introduction is generally very well written and has most of the relevant points addressed. Some extra points that will improve it and increase its value are:
- Adding a short part that presents all different phages found to infect oenococci
- Explaining which are the overlapping niches between O. oeni and its sister species (line 48)
- Give one or more e.g. of interesting questions raised (line 62)
- The last paragraph has many elements that belong to a text written for methods and results. Replace it with a clear statement of your motivation or aim for conducting this study.
III. Materials and Methods
This section requires considerable improvements. As a general comment, please read though the “Results and Discussions” section and collect all the methods referenced there that are absent from the text. Moreover, when introducing a new method in this section please consider shortly explaining what this method was used for. For e.g. “ To further assess any potential interactions between strains of Oenococcus sp. and phages infecting oenococci, we ….”
Last but not least for reproducibility and clarity reasons please collect the names, source and accession numbers (where available) of all the plasmids, bacteria and phages named in this study and provide either as a Table in this section or as supplementary material. Specifically for the phages it is crucial that the authors provide the accession numbers and make available any missing genomes, as the bacteriophage genomes are the baseline of this study.
The following comments all boil down to providing a manuscript with a clearly reproducible study where all methods are well described at the Materials and Methods section.
- Please clarify how you have performed the double-agar overlay technique (lines 89-90). Was the lysate tested using a spotting assay or by addition to the mix of agarose and overnight culture? If it was done by spotting you risk seeing plaques that are the results of lysis from without. Moreover, which exactly was the indicator strain used? You mention an indicator strain on the “Results and Discussions” section but I would advise you to include this information here. The reference nr. 16 provided here is not relevant to the way you conducted this experiment so please remove it and include instead the aforementioned details for further clarity.
- Mention which were the hosts used.
- Please add the results of lines 92-96 to the “Results” section.
- In line 104 please add the specific settings you used to run CRISPRCasFinder.
- In line 107 you have given n=4 for your plasmids but later on (lines 242-244) you mention that you used 5 plasmids. Also in line 107 you mention lysogenic bacteria but the number is missing. Please check this sentence and correct if relevant.
- Line 108-110: Further information on how the manual comparisons were performed is needed here.
IV. Results and Discussion
The Results and Discussion section presents and in general properly discusses the very exciting findings of this study.
The most crucial improvements that should be made to this section are: a) to remove of all the elements that belong to the material and methods section and transfer the ones that are missing from the M&M section there, and b) to ensure all data that can ensure reproducibility, transparency and fairness of the study are made available.
- Figure 1A is taken without any adaptations from reference nr. 9. It is my strong recommendation that the authors remove the Figure 1 A and just reference the study where this is presented, as this is material that has been published before and could be mistaken for plagiarism if read quickly.
-Line 125: You refer to some results that could be added to Figure 1B. Please consider this option.
-Figure 1S: The method and software used for constructing this tree is missing and should be described in “Materials and Methods”. You also need to add one more phrase that connects the results from BLAST with the results shown in the phylogenetic tree of Figure S1.
- Line 152: The description of how the pairwise genomic comparisons are done is missing from the Material and Methods section, and should be added there. Also in the same line, please specify which the reference strain you are referring to is.
- Line 157: There is a reference missing. Please see specifics on the pdf comment.
- Lines 155-157: The results presented are the outcome of annotation analysis. In one case HHPred is reference but the analysis overall is missing from the M&M section. It is essential that the authors fully describe the annotation analysis there. Also it is crucial that the authors publish the sequences of all the hypothetical genes and how these were annotated by this study for reproducibility purposes. Should the hypothetical gene sequences already exist in one of NCBI’s nucleotide database it is sufficient to provide the NCBI ID.
-Line 162: Please correct to 29-36 nucleotides based on what you present in Table1.
- Line 164: Please provide the sequences of the DR that are similar. This is a very interesting observation and it may be that among these DR sequences there are some sequences that are favoured within LAB but if not provided no researcher outside the authors will be able to investigate this further now or as more LAB genomes become available.
- Table S1 and Table S2 are missing. Please provide them.
- The last paragraph of 3.1 requires some rearrangements. Please find the specific comments in the pdf document.
- Section 3.2 does not fit very well with the rest of the Results and Discussion section, partially because it includes a lot of methodology elements. Thus 3.2 requires restructuring. Please move those elements to the methods section and only refer to results of your typing and selection of representative phage genomes here. It seems to me that the part that is clearly referring to results is starting around line 211.
- Line 208: Here you refer to another round of annotation efforts that is completely missing from your M&M section. Please introduce it there. Was the annotation only done using RAST or did it also involve manual curations, which would make the annotations more robust & trustworthy? Also the gene sequences and their annotations should be provided for reproducibility as already requested in comment of lines 155-157 above.
-Line 212: In line 208 you refer to three main categories but here you mention eight. Please refrain from using numbers and just use the categories by their names. That would make the text easier to follow :)
- I suggest that the part of lines 235 – 241 (starting with “ As shown”) rather belongs to the section that discusses the protospacers and where Figure 4 (erroneously referenced as Figure 3 in line 235) is extensively presented. Please consider moving there.
- Line 243: Here you refer to the plasmids included in this study’s database. Please explain (preferentially in the M&M section) why you choose only 5 plasmids for this study’s database, just as you describe why you chose 37 representative phages. There are more plasmids of O. oeni that have been described (see as e.g. paper with doi: 10.1006/plas.1999.1397)
- In lines 253-255 the authors claim that many more protospacers are found in lytic phages of O. oeni as compared to plasmids and prophages. However, the database used contains many more prophages compared to lytic phages and plasmids, so any direct comparisons are not possible! It would strongly advise the authors to remove this sentence.
- In lines 262-3 you are discussing that the abundances of the eleven spacers are inversely correlated with phylogenetic distances among O. oeni and its congeneric species. However, this is not really true for the plasmids, so you would need to correct the sentence to only refer to the phages. The fact that this observation is not true for the plasmids may be due to the fact that your database includes very few plasmids to draw any robust conclusions. Please provide your thoughts regarding these.
- Table 2: the results of the other 2 plasmids (pOeni1 and pAWRIB429)are missing from the table, please add. Please also see a suggestion of how to modify the title of the table in the attached pdf. Lastly, you include some hypothetical proteins (HP) as functions targeted which are later in the text annotated (e.g. the last HP is later referred to as Tgase in Figure 3 and paragraph 3.4). Please add these annotations to the table and make all sequences of the annotated genes available as earlier requested.
- Lines 280-2: Another part that needs to be properly described in M&M.
- Lines 301-2: The method is missing from M&M.
- Lines 305-6: Here and again in lines 330-7 you refer to analyses performed to the mentioned prophage. However, these analyses are missing from the M&M. For e.g., how can you prove that the lysate only contains one prophage? And how did you identify the integration site and PAM? Please properly describe the methods you have used related to this prophage. Moreover, please make the sequence of the prophage available for reproducibility purposes.
- Please ensure that Figures 3-5 are of high quality when submitted and that everything is readable.
- Line 330: Cas3 PAMs are normally located upstream the protospacer. How do the authors explain that this PAM is predicted to be downstream?
- Line 345: Method not mentioned in M&M, please add.
- Line 351: Do the authors have any further thoughts regarding why the phages of Cluster II.1 are not targeted? Could the environment where these phages come from be a reason?
- Line 380: Method not mentioned in M&M, please add.
- Fig.5A: Information for SPOSI89 is not included.
- Line 388: It is not clear what your message is here, please elaborate. Why do you think this is happening?
- Section 3.6: Please explain the criteria you used to select the different indicator strains here. Is it based on phylogenetic distances? Also how did you ensure that your lysates mentioned in line 417 includes only one induced prophage and not more?
V. Conclusions
- Line 434: Your claim here cannot be supported by the dataset of this study, since it includes only 5 plasmids. Please remove.
VI. References
- Please remove reference 29. I believe that posters do not qualify for references.
Further minor comments
Minor comments are marked in the pdf attached. Comments in yellow are correcting language and expression errors.

Reviewer 2 Report
The manuscript details the analysis of the CRISPR-Cas systems and spacers to define the history of phage-host interactions in the genus at species and cross-species levels. It is a highly understudied organism and therefore this study represents a significant advance in understanding of the systems present, their diversity and the historical interactions of phages in the genus. I have presented minor comments to be addressed below.
L21: beverages
L28: Why italicise Grapes? Similar in L30, L78, L120-2 and other places in the manuscript where some text is italicised without apparent reason. Please check and correct as appropriate.
L86: Please indicate at what OD600 the mitomycin C was added to the culture to cause induction.
L106-8: It is not clear where the database comes from- how was it constructed? A little more detail would be very helpful to the reader.
L139: Lactobacilli does not need to be italicised.
L147: Italicise Oenococcus
L336: The finding of the ACRs is very interesting. It would be beneficial to the reader perhaps to indicate the function/specificity of these systems (which CRISPR/Cas systems and genes do they target)
The authors omitted funding acknowledgements section. Insert if appropriate.
Author Response
see the unique attachment
